# Partial Substitution of Fermented Soybean Meal for Soybean Meal Influences the Carcass Traits and Meat Quality of Broiler Chickens

**DOI:** 10.3390/ani10020225

**Published:** 2020-01-30

**Authors:** Shuangshuang Guo, Yuanke Zhang, Qiang Cheng, Jingyun Xv, Yongqing Hou, Xiaofeng Wu, Encun Du, Binying Ding

**Affiliations:** 1Engineering Research Center of Feed Protein Resources on Agricultural By-products, Ministry of Education, Hubei Key Laboratory of Animal Nutrition and Feed Science, Wuhan Polytechnic University, Wuhan 430023, China; guo1shuangshuang@163.com (S.G.); zhangyuanke163@163.com (Y.Z.); chengqiangcool@163.com (Q.C.); xvjingyun@163.com (J.X.); houyq@aliyun.com (Y.H.); 2Hubei (Wuhan) Broad Live-Stock Technique Co., Ltd., Wuhan 430071, China; whbzd@whbzd.com.cn; 3Hubei Key Laboratory of Animal Embryo and Molecular Breeding, Institute of Animal Science and Veterinary Medicine, Hubei Academy of Agricultural Sciences, Wuhan 430064, China; qiaowan77@126.com

**Keywords:** broiler chicken, fermented soybean meal, growth performance, carcass trait, meat quality

## Abstract

**Simple Summary:**

Fungal and bacterial fermentation improves the nutritional quality of soybean meal (SBM). The beneficial effects of fermented soybean meal (FSBM) on growth performance and gut health of broiler chickens have been demonstrated. However, FSBM is usually used in pre-starter diets of broiler chickens due to the high cost. In the present study, the SBM in diets was partially replaced by FSBM to evaluate its effect on the growth performance, carcass traits, and meat quality of broiler chickens. The growth performance and major carcass traits were not significantly affected by FSBM treatments. Different levels (2.5%, 5.0%, and 7.5%) of FSBM supplementation influenced the meat color, pH, nutritional composition, and antioxidant properties. The improvement of meat quality would extend the shelf life of meat and increase consumer acceptability to chicken. The 2.5% FSBM was recommended in a broiler diet.

**Abstract:**

The usage of fermented soybean meal (FSBM) in poultry feed is limited due to the high cost. The present study was conducted to examine the carcass traits and meat quality of broiler chickens that were fed diets with partial replacement of soybean meal (SBM) with FSBM. The 336 one-day-old chicks were assigned to four groups with 0% (control), 2.5%, 5.0%, and 7.5% FSBM addition in corn-SBM-based diets. Compared with the control, 2.5% and 5.0% FSBM decreased leg muscle yield, breast drip loss, and cooking loss (*p* < 0.05). The 7.5% FSBM increased the ultimate pH of breast and thigh muscles, and all FSBM treatments decreased muscle lightness and breast malondialdehyde content (*p* < 0.05). The 2.5% FSBM increased breast total superoxide dismutase activity, while 7.5% FSBM reduced breast hydrogen peroxide level (*p* < 0.05). All FSBM treatments elevated breast contents of bitter and sour tasting amino acids, and 2.5% and 7.5% FSBM increased breast glutamic acid and total free amino acids (*p* < 0.05). The 5.0% and 7.5% FSBM elevated thigh isoleucine and leucine contents (*p* < 0.05). In conclusion, FSBM replacing SBM affected meat quality with the decrease of lightness and increase of pH, water-holding capacity, antioxidant properties, and free amino acids.

## 1. Introduction

Soybean meal (SBM) is the most commonly used dietary protein in the poultry and swine feed industries. The desolventizing–toasting process during the production of SBM greatly decreased the anti-nutritional factors in raw beans to low levels, but residual anti-nutritional compounds, such as trypsin inhibitors, lectins, and soya globulins, limited its application in piglets and post-hatch chicks [1,2]. Fungal and bacterial fermentation could improve the nutritional quality of SBM [3]. After a bed-packed solid fermentation of SBM with *Aspergillus oryzae* GB-107 for 48 h, the 84% of trypsin inhibitor was eliminated, and 10% more crude protein, and 62.5% more small-size peptides (<20 kDa) were produced in fermented SBM (FSBM) [4]. Solid fermentation of SBM with *Bacillus subtilis* XZI125 reduced the trypsin inhibitor, phytic acid, and urease content by 33.57%, 42.65%, and 67.31%, respectively, and yielded 171.32 mg/g of acid-soluble peptides [5].

It was reported that totally substituting FSBM fermented by *A. oryzae* 3.042 for SBM in a broiler chicken diet significantly increased average daily gain (ADG) and average daily feed intake (ADFI) in both starter and grower phases, enhanced the activities of intestinal trypsin, lipase, and protease of starter broilers, as well as the protease activity of grower broilers, elevated villus height, and decreased crypt depth of jejunum mucosa in the overall phase [6,7]. Furthermore, investigation demonstrated that feeding broiler chickens with FSBM fermented by *Bacillus* alone, or together with a yeast by-product at the expense of 3% dehulled SBM in the first 7 days post-hatch, increased ADG and decreased feed conversion ratio (FCR) during total rearing period, elevated jejunal villus height at d 7, increased lactic acid bacteria and *Bacillus* spp., and reduced *Coli*-form bacteria in cecal contents at d 35 [2]. Although FSBM showed improvements in growth performance and intestinal health of broiler chickens, its effects on carcass traits and meat quality were largely unknown. 

Meat is a complex, composite substance. It consists of myofibers, connective tissue, and lipids. The pH is one of the most important parameters for meat quality, as it has a positive correlation with the water holding capacity, as well as redness and tenderness [8], and a negative correlation with the lightness [9] and drip loss of meat [10]. In the early postmortem period (24 h postmortem), pH declines rapidly due to the anaerobic glycolysis and lactic acid accumulation [11]. The reduction of pH leads to the extensive denaturation of protein, which in turn affects the color and water-holding capacity of meat [12,13]. Raw chicken meat is generally very soft and, when cooked, it can even be cohesive [14]. Therefore, tenderness is another important meat quality trait and is reflected by shear force.

The influences of fermented feed on the carcass traits and meat quality of pigs and poultry have been documented. Fermented biomass residue did not significantly affect the meat quality of pigs, but increased the essential, non-essential, and total amino acids in pork [15]. Fermented rapeseed cake had no effect on carcass dressing percentage or carcass fat content, but decreased the yield of breast muscle and increased the pH at 24 h postmortem, as well as the antioxidant status in breast muscle of turkey [16]. Furthermore, fermented products in animal feed are considered as an alternative to antibiotics due to their probiotic properties [17]. However, the inclusion of FSBM in a broiler chicken diet is limited because of the high cost [2]. Therefore, three low-substituting levels of FSBM for SBM were designed to investigate their effects on growth performance, carcass traits, and meat quality of broiler chickens.

## 2. Materials and Methods

### 2.1. Experimental Birds and Diets

All animal procedures were performed in compliance with China’s guidelines for the Review of Laboratory Animal Welfare and Ethics (GB/T 35892-2018), and were approved by the Institutional Animal Care and Use Committee of Wuhan Polytechnic University (20180406). A total of 336 one-day-old chicks (Ross 308, Xiangyang Charoen Pokphand Co., Ltd., Hubei, China), hatched from the same breeder flock, and supplied by a commercial hatchery, were housed in a poultry house, weighed, and randomly allocated into 4 treatment groups with 7 replicate pens of 12 birds (6 males and 6 females), based on a randomized complete block design. The birds were housed in wire cages (850 cm^2^/bird) in an environmentally controlled room with 23-h light, and were allowed ad libitum access to water and feed. The room temperature was initially set at 33 °C and gradually decreased by 4 °C weekly until a final temperature of 23 °C was reached. The starter (d 1–21) and grower (d 22–42) diets were formulated according to the nutritional requirements of the National Research Council [18]. The four experimental diets consisted of a control corn-SBM diet and three test diets in which the SBM was partially replaced with 2.5%, 5.0%, and 7.5% of FSBM, respectively. All diets were formulated with equal metabolizable energy and crude protein levels, and produced in the form of mash. The FSBM, which was fermented by *B. subtilis*, *Lactobacillus* spp., and yeasts, was provided by Hubei (Wuhan) Broad Live-Stock Technique Co., Ltd. (Wuhan, China). The analytic composition of FSBM is shown in Table 1. All diets contained similar amounts of crude protein, major amino acids (including lysine, methionine with cysteine, threonine), minerals (including calcium and available phosphorus), and vitamins. The ingredient composition of diets is shown in Table 2. The calculated and analyzed nutrient levels of experimental diets are presented in Table 3. The analysis of nutrient composition in diets was performed according to the procedures of the Association of Official Analytical Chemists [19]. 

### 2.2. Growth Performance

The broiler chickens in each replicate were weighed at d 1, 21, and 42. The feed consumption per replicate was recorded by subtracting the remaining feed weight from initial feed weight during starter and grower phases. The ADG, ADFI, FCR, and mortality was further calculated within starter, grower, and overall phases (d 1–42). The dead birds were excluded when the growth performance was evaluated.

### 2.3. Sample Collection

On d 42 of the trail, 2 birds (1 male and 1 female) from each replicate (14 birds per group) were randomly selected following a 12-h feed withdrawal, weighed individually, euthanized by cervical dislocation, and then were immediately bled. After defeathering, the head removal, hock cut, and evisceration was performed, and the hot carcass weight was recorded. The carcass yield was calculated by dividing final body weight with hot carcass weight. The abdominal fat, breast (pectoralis major and minor), and leg (thigh and drumstick) muscles were removed and weighed. Their respective percentages based on hot carcass weight were calculated. Approximately 5.0 g of muscles from the medial pectoralis major and thigh were sampled for measurement of antioxidative status and free amino acids. The remaining muscles from each bird were kept at 4 °C for the analysis of meat quality.

### 2.4. Meat Quality Measurements

The ultimate pH of breast and thigh muscle was measured at 24 h postmortem by insertion of a pH meter (Ingold, Mettler Toledo, Greifensee, Switzerland) equipped with a penetrating glass electrode. The pH meter was calibrated by measuring buffer solutions at pH 4.0 and 7.0 at ambient temperature. Meat color attributes including lightness (L*), redness (a*), and yellowness (b*), were measured (average value of 3 measurements was taken from the middle and 2 corners of the muscle samples) using Chroma Meter (Opto-Star Lab, Matthaus, Germany), according to the CIELAB trichromatic system (CIE, Commission Internationale de I’Eclairage, Vienna, Austria). The measurements of shear force and drip loss were conducted as described by Liu et al. [20]. Approximately 20 g of a sample from the breast and thigh muscles was heated in a water bath at 85 °C until the internal temperature reached 80 °C. After cooling to room temperature, the sample was cut into 1 cm × 1 cm × 3 cm along the direction of the muscle fibers. The shear force was measured using a digital texture analyzer (Model 2000D, G-R, US). Approximately 5 g of muscle was weighed and put into zip-lock plastic bags, hung from a hook, and left at 4 °C for 24 h. Then, the surface moisture of the sample was absorbed with filter paper and the sample was reweighed. According to the method of Alnahhas et al. [21], approximately 30 g of muscle was weighed, wrapped in an airtight polythene bag, and cooked in a water bath at 85 °C for 13 min (internal temperature 70 °C). Afterwards, the samples were cooled in crushed ice for 10 min, wiped dry using absorbent paper, and weighed. Both drip loss and cooking loss were expressed as a percentage of the fresh muscle weight. 

### 2.5. Assay of Muscle Antioxidative Status 

Approximately one gram of breast or thigh meat was homogenized in 10 mL of ice-cold saline and centrifuged (3000× *g*, 10 min, 4 °C). The supernatants were collected for further analysis. The activities of total superoxide dismutase (T-SOD), as well as the contents of malondialdehyde (MDA), and hydrogen peroxide (H_2_O_2_) in the supernatants were determined using commercial kits purchased from Nanjing Jiancheng Institute of Bioengineering (Nanjing, China), according to the manufacturer’s instructions [22].

### 2.6. Measurement of Free Amino Acids

The free amino acid determination procedures were performed according to the methods described by Perenlei et al. [23] with modifications. Briefly, the frozen muscle samples were weighed (approximately 1.0 g) and after adding 25 mL of 2% salicyl-sulphonic acid were homogenized thoroughly. The homogenate was centrifuged at 10,000× *g* for 15 min to obtain the supernatants, which were diluted 20 times, and filtered through a polyvinylidene fluoride membrane filter before analysis. The free amino acids in the extracts were measured using an automatic amino acid analyzer (S433D, Sykam GmbH, Eresing, Germany). The contents of free amino acids were expressed as μg/g wet tissue. The amino acids were classified as sweet, bitter, sour, and Umami-tasting, according to the description of Perenlei et al. [23].

### 2.7. Statistical Analysis

All the data were analyzed with SPSS version 21.0 (SPSS Inc., Chicago, IL, USA). Results are expressed as the mean ± pooled SEM. The data were analyzed by one-way analysis of variance. For growth performance, the individual cage was considered an experimental unit; for slaughter performance and meat quality, the individual bird was considered an experimental unit. Differences among means were tested by Duncan’s test. Significance was declared at *p* < 0.05.

## 3. Results

### 3.1. Growth Performance

As shown in Table 4, compared with the control, FBSM ranging from 2.5% to 7.5% did not significantly affect the growth performance (ADG, ADFI, FCR, and mortality) of birds in starter, grower, and overall periods. 

### 3.2. Carcass Traits

The carcass traits of broiler chickens are presented in Table 5. The final body weight, hot carcass weight, carcass and breast yields, and abdominal fat percentage were not significantly influenced by the dietary treatments (*p* > 0.05). However, 2.5% and 5.0% FBSM groups had lower leg muscle yields than that of control (*p* < 0.05), while the 7.5% FSBM group had intermediate leg muscle yield (*p* > 0.05).

### 3.3. Meat Quality

As summarized in Table 6, 7.5% FSBM group had higher pH of breast muscle than that of the other three groups (*p* < 0.05), and among the latter, breast pH did not differ (*p* > 0.05). Further, the pH value of the thigh muscle was higher in the treatment with FSBM at 7.5%, compared to the values from the other groups (*p* < 0.05). The lowest pH values were found in the control and 5.0% FSBM groups (*p* > 0.05), with an intermediate pH value in the 2.5% FSBM group (*p* < 0.05).

Compared with the control, all FSBM treatments decreased the lightness (L*) of both breast and thigh muscle (*p* < 0.05). Comparison among FSBM treatments showed that the 5.0% FSBM group had lower breast lightness than that of the other two groups (*p* < 0.05), while the thigh showed lower lightness in the 2.5% and 5.0% FSBM groups, compared to the 7.5% FSBM group (*p* < 0.05). The 5.0% FSBM treatment decreased the redness (a*) and yellowness (b*) of breast muscle compared with the other three groups (*p* < 0.05). The 5.0% FSBM tended to reduce the redness of the thigh muscle compared with the control and 7.5% FSBM group (*p* = 0.085). Yellowness of the thigh muscle was similar among the groups (*p* > 0.05).

Only the shear force of the thigh muscle was significantly affected by treatments. In fact, thigh meat from the 5.0% FSBM group was tougher compared to that of the 7.5% FSBM group (*p* < 0.05); higher shear force value was found also in the 2.5% FSBM group than that in the control and 7.5% FSBM groups, but the differences were not significant (*p* > 0.05). In contrast to the control, both 2.5% and 5.0% FSBM reduced the drip loss and cooking loss of breast muscle (*p* < 0.05), while 7.5% FSBM did not affect these indices (*p* > 0.05). Drip loss and cooking loss of thigh were not affected by treatments (*p* > 0.05).

### 3.4. Muscle Antioxidative Status

As seen in Table 7, treatment with 2.5% FSBM increased the activity of T-SOD in breast muscle compared to the control (*p* < 0.05), while 5.0% and 7.5% FSBM had intermediate breast T-SOD activity (*p* > 0.05). In contrast to the control, all FSBM treatments decreased the breast MDA contents (*p* < 0.05) and the 7.5% FSBM group had a lower breast MDA level than that of the other two FSBM groups (*p* < 0.05). The 7.5% FSBM decreased the breast H_2_O_2_ content compared with the other groups (*p* < 0.05), with no significant differences among the latter. The 7.5% FSBM group had a higher thigh H_2_O_2_ level than that of the control and 5.0% FSBM groups (*p* < 0.05), with intermediate thigh H_2_O_2_ content in the 2.5% FSBM group (*p* > 0.05). The thigh T-SOD activity and MDA content was not influenced by treatments (*p* > 0.05).

### 3.5. Free Amino Acids in Muscle

As presented in Table 8, the contents of free amino acids in breast muscle were greatly affected by dietary FSBM treatments. As to the sweet tasting amino acids, 2.5% and 7.5% FSBM groups had higher levels of alanine and phenylalanine in breast muscle than those of control and 5.0% FSBM groups (*p* < 0.05). All FSBM groups decreased the breast glutamine content in comparison with the control (*p* < 0.05), and the 2.5% FSBM group had lower value than those of the other two FSBM groups (*p* < 0.05). The 5.0% FSBM reduced breast glycine level compared to other groups (*p* < 0.05). The control and 2.5% FSBM groups had the lowest and highest levels of lysine in breast muscle, respectively (*p* < 0.05); the 5.0% and 7.5% FSBM groups had intermediate levels of lysine (*p* < 0.05). Significant differences (*p* < 0.05) in serine contents were found between control and the groups 2.5% and 7.5% of FSBM; all FSBM groups showed different serine contents (5.0% < 7.5% < 2.5%; *p* < 0.05). All FSBM treatments increased breast threonine contents in comparison to the control (*p* < 0.05); threonine levels were different among the FSBM groups with 2.5% > 7.5% > 5.0% (*p* < 0.05). The thigh showed a significant difference only in serine content that was higher in the 2.5% FBSM group compared with the other groups (*p* < 0.05).

Compared to the control, all FSBM groups increased the contents of bitter tasting amino acids (arginine, histidine, isoleucine, leucine, methionine, and valine) in breast muscle (*p* < 0.05). The breast bitter tasting amino acids differed among the FSBM groups, with 2.5% > 7.5% > 5.0% (*p* < 0.05). Isoleucine and leucine contents in the thigh muscle increased in the 5.0% and 7.5% FSBM groups, compared to the control (*p* < 0.05), and intermediate contents were found for the 2.5% group (*p* > 0.05). Methionine contents in the thigh muscle were higher in FSBM groups compared to the control group; however, only 7.5% resulted in a significantly higher value (*p* < 0.05).

Compared with the control, all FSBM treatments elevated breast level of aspartic acid, a sour-tasting amino acid (*p* < 0.05), and 2.5% and 7.5% FSBM groups had higher breast aspartic acid content than that of the 5.0% FSBM group (*p* < 0.05). The 2.5% and 7.5% FSBM groups had higher breast content of glutamic acid (a Umami-tasting amino acid), and total free amino acids, than that of the control (*p* < 0.05), and the 5.0% FSBM group exhibited similar values to the control (*p* > 0.05). The levels of sour and Umami-tasting amino acids, as well as total free amino acids in the thigh muscle, were not affected by treatments (*p* > 0.05). 

## 4. Discussion

It was demonstrated that FSBM improved the growth performance of broiler chickens with total substitution for SBM [6,24]. In the present study, partial FSBM substitution did not significantly affect the growth performance of birds. Mathivanan et al. [25] reported that dietary inclusion with 0.5% FSBM fermented by *Aspergillus niger* increased body weight in the fifth and sixth week, and decreased FCR in the third and fourth week, but the addition of 1.0% and 1.5% FSBM did not show beneficial effects on the growth performance of broiler chickens. A trial with fungi and bacteria-mixed fermented SBM showed that 4.5–6.0% wet or dry FSBM did not show superior effects on growth performance of broiler chickens, compared with 4% fish meal, but exhibited better effects than that of 5% SBM [26]. The inconsistent results between our findings and the previous observations might be attributed to the different microbes used for fermentation, the different processing methods for FSBM products, or the different supplemented levels of FSBM in the diet. 

Consistent with the unaffected growth performance, the hot carcass weights and percentages of breast muscle and abdominal fat were not influenced by FSBM treatments in the present study. Similarly, Kim et al. [2] reported that dietary supplementation with 3% FSBM products during 7 d after hatch did not affect the relative weights of both breast muscle and abdominal fat at d 35. Moreover, partial substitution of FSBM for SBM (4.5–6.0%) did not change the weight and relative weight of abdominal fat in broiler chickens at d 37 [26]. However, the decrease of leg muscle yield in 2.5% and 5.0% FSBM treatments needs further investigation. 

In the present study, 7.5% FSBM increased the ultimate pH and reduced the lightness of both the breast and thigh muscle. The pH values detected in the 7.5% FSBM group were higher (6.69 and 6.60 in breast and thigh, respectively) than that reported in the literature of broiler chickens slaughtered at 42 days of age (pH 5.82–5.87 in Ross [14]; pH 5.67–5.99 in Cobb [27]). This could be due to the stress that birds in the 7.5% FSBM group may have sustained prior to slaughter, which could have resulted in the depletion of glycogen in the muscle [28]. Since glycogen is the substrate for lactate production in muscle, the less glycogen that is present at harvest, the less lactate is produced after harvest, and subsequently, the less the pH will decline in postmortem muscle. Differently, the pH values observed in 2.5% and 5.0% FSMB groups, and the control group, were fully fit within the pH range (5.40–5.99) accepted for commercial poultry meat [29].

FSBM treatments did not affect the meat tenderness compared with the control, but thigh meat in the 7.5% FSBM group was much more tender than that in the 5.0% FSBM group. It was reported that meat tenderness negatively correlated with myofiber density and thickness of the endomysium, and positively correlated with thickness of the perimysium [30]. However, the myofiber density of both the breast and thigh muscle was not significantly influenced by FSBM treatments in the present study (data not shown). The muscle membrane characteristics need further investigation. Both drip loss and cooking loss could reflect the water-holding capacity and there was a positive correlation between them [31]. The 2.5% and 5.0% FSBM decreased both drip loss and cooking loss of breast muscle, which suggested the increase of water-holding capacity of meat. There is little literature focused on the effects of FSBM on meat quality of animals. 

In the current study, FSBM treatments promoted the antioxidative properties of the breast muscle by increasing antioxidant enzyme activity and inhibiting lipid peroxidation and peroxide accumulation. This improvement of antioxidative capacity would benefit the meat quality and shelf life [32]. The antioxidant function of FSBM might be derived from its fermentation products. It was reported that during the fermentation of SBM with *Bacillus amyloliquefaciens* SWJS22, the antioxidant components, such as the phenolic and flavonoid compounds, increased sharply, and the intragastric administration of lyophilized FSBM supernatant improved the activities of T-SOD, glutathione peroxidase (GSH-Px), catalase and total antioxidant capacity (T-AOC), and inhibited the formation of MDA in serum and liver of mice [33]. The fermented feed has shown improved antioxidative status in broiler chickens. Dietary addition with 10% fermented wheat bran replacing maize up-regulated the antioxidant gene expression in chicken peripheral blood mononuclear cells [34]. Supplementation with cultured media of solid-state fermented *Isaria cicadae* in a diet increased the T-AOC, GSH-Px, and T-SOD activities in the serum and kidney of broiler chickens with dose-dependent manner [35]. However, little is known about the effects of fermented feed on the redox state of meat. 

Free amino acids, especially glutamic acid, can greatly contribute to the taste of meat [36]. Both total free amino acids and glutamic acid in breast muscle was increased by 2.5% and 7.5% FSBM treatments in the present study. Furthermore, all bitter and sour tasting amino acids, as well as sweet tasting amino acids (lysine and threonine), in breast muscle were elevated by all FSBM treatments. In the thigh muscle, less free amino acids were significantly affected by FSBM treatments, and only the serine, isoleucine, leucine, and methionine contents were increased. The different responses of breast and thigh muscles to FSBM treatments might be due to the higher contents of total amino acids in the breast muscle, which made it more sensitive. Chen et al. [37] reported that breast muscle contained higher levels of total essential and non-essential amino acids than those of thigh muscle in commercial broilers. It was demonstrated that the growth of breast muscle required higher ideal ratios of digestible methionine and threonine than leg muscle in boilers [38], which indicated the inconsistent responses of breast and leg muscles. Free amino acids usually increase during the postmortem storage of meat due to the action of aminopeptidases and proteases [39]. These enzymatic reactions in proteolytic system are affected by meat pH reduction [40]. In the present study, 7.5% FSBM significantly increased the ultimate pH of both breast and thigh muscle, and the other two FSBM treatments elevated meat pH with or without significance. This might partly contribute to the increase of free amino acids in FSBM treatments. Moreover, the meat antioxidant status influences the contents of free amino acids. Higher antioxidative capacity would inhibit the oxidation of enzymes and amino acids [23]. The improvements of antioxidant status of breast muscle by FSBM treatments were another factor resulting in the accumulation of free amino acids in breast muscle. 

## 5. Conclusions

Partially substituting FSBM for SBM did not affect the growth performance of broiler chickens, but influenced the meat quality by decreasing lightness and increasing the ultimate pH, water-holding capacity, antioxidant properties, and free amino acids. The 2.5% FSBM was recommended in a broiler diet to improve meat quality, which might extend the shelf life of meat and increase the consumer acceptability to chicken. Nevertheless, further research is needed to increase knowledge regarding the effect of partial replacement of SBM with FSBM on the growth and meat quality of broiler chickens.

## Figures and Tables

**Table 1 animals-10-00225-t001:** Analytic nutrient composition of fermented soybean meal (FSBM).

Item	Nutrient Composition (%)
Dry matter	88.0
Crude protein	50.1
Lysine	3.56
Methionine	0.55
Threonine	1.90
Calcium	0.34
Total phosphorus	0.67
Probiotics (CFU/g)	≥1.0 × 10^6^

**Table 2 animals-10-00225-t002:** Ingredient composition of experimental diets.

Ingredients (%)	Starter Diets (d 1–21)					Grower Diets (d 22–42)			
	Control	2.5% FSBM	5.0% FSBM	7.5% FSBM		Control	2.5% FSBM	5.0% FSBM	7.5% FSBM
Corn	51.48	52.21	52.95	53.68		57.56	58.30	59.03	59.76
Soybean meal	40.78	37.64	34.49	31.35		35.15	32.00	28.85	25.71
Soybean oil	3.44	3.34	3.23	3.13		3.66	3.56	3.46	3.36
Fermented soybean meal (FSBM)	0.00	2.50	5.00	7.50		0.00	2.50	5.00	7.50
Dicalcium phosphate	1.92	1.93	1.94	1.95		1.33	1.34	1.35	1.36
Limestone	1.16	1.16	1.16	1.16		1.26	1.26	1.26	1.26
Sodium chloride	0.35	0.35	0.35	0.35		0.35	0.35	0.35	0.35
DL-Met (98%)	0.26	0.26	0.26	0.26		0.13	0.13	0.13	0.13
L-Lys (78%)	0.00	0.00	0.01	0.01		0.00	0.00	0.01	0.01
Choline chloride (50%)	0.25	0.25	0.25	0.25		0.20	0.20	0.20	0.20
Trace mineral1	0.20	0.20	0.20	0.20		0.20	0.20	0.20	0.20
Vitamin premix2	0.05	0.05	0.05	0.05		0.05	0.05	0.05	0.05
Sodium propionate (99%)	0.05	0.05	0.05	0.05		0.05	0.05	0.05	0.05
Ethoxyquin (66%)	0.03	0.03	0.03	0.03		0.03	0.03	0.03	0.03
Zeolite powder	0.03	0.03	0.03	0.03		0.03	0.03	0.03	0.03

The trace mineral premix provided the following (per kilogram of diet): copper, 8 mg; zinc, 75 mg; iron, 80 mg; manganese, 100 mg; selenium, 0.30 mg; iodine, 0.35 mg. The vitamin premix supplied the following per kilogram of complete feed: vitamin A (retinyl acetate), 12,500 IU; vitamin D3, 2,500 IU; vitamin K3, 2.65 mg; vitamin B1, 2 mg; vitamin B2, 6 mg; vitamin B12, 0.025 mg; vitamin E (DL-α-tocopheryl acetate), 30 IU; biotin, 0.0325 mg; folic acid, 1.25 mg; pantothenic acid, 12 mg; niacin, 50 mg.

**Table 3 animals-10-00225-t003:** The calculated and analyzed nutrient levels of diets.

Nutrient Levels (%, Unless otherwise Indicated)	Starter Diets (d 1–21)					Grower Diets (d 22–42)			
	Control	2.5% FSBM	5.0% FSBM	7.5% FSBM		Control	2.5% FSBM	5.0% FSBM	7.5% FSBM
Calculated									
Metabolizable energy (Mcal/kg)	2.92	2.92	2.92	2.92		3.00	3.00	3.00	3.00
Crude protein	21.50	21.50	21.50	21.50		19.50	19.50	19.50	19.50
Calcium	1.00	1.00	1.00	1.00		0.90	0.90	0.90	0.90
Available phosphorus	0.45	0.45	0.45	0.45		0.35	0.35	0.35	0.35
Lysine	1.17	1.17	1.17	1.17		1.04	1.04	1.04	1.04
Methionine + Cystine	0.90	0.90	0.90	0.90		0.72	0.72	0.72	0.72
Threonine	0.82	0.82	0.82	0.82		0.74	0.74	0.74	0.74
Analyzed									
Crude protein	21.6	21.8	21.7	21.6		20.5	20.9	20.8	20.2
Calcium	1.04	1.00	0.96	1.02		0.85	0.97	0.87	0.94
Total phosphorus	0.74	0.71	0.69	0.73		0.59	0.62	0.58	0.59
Lysine	1.17	1.18	1.15	1.24		1.17	1.21	1.21	1.27
Methionine	0.50	0.48	0.46	0.47		0.38	0.37	0.39	0.35
Threonine	0.74	0.74	0.72	0.78		0.76	0.77	0.75	0.77

**Table 4 animals-10-00225-t004:** Effects of fermented soybean meal (FSBM) on growth performance of broiler chickens.

Item	Dietary Treatment				SEM	*p*-Value
	Control	2.5% FSBM	5.0% FSBM	7.5% FSBM		
Starter (d 1–21)						
ADG (g/bird per day)	34.01	34.42	34.33	33.25	0.32	0.57
ADFI (g/bird per day)	50.06	51.01	50.34	50.45	0.49	0.928
FCR (g/g)	1.473	1.482	1.467	1.518	0.011	0.346
Mortality (%)	1.19	2.38	1.19	2.38	0.66	0.861
Grower (d 22–42)						
ADG (g/bird per day)	73.72	74.42	70.64	71.44	0.81	0.307
ADFI (g/bird per day)	134.1	132.98	128.56	131.71	1.51	0.615
FCR (g/g)	1.822	1.792	1.823	1.846	0.023	0.892
Mortality	4.29	1.59	1.43	1.59	0.97	0.701
Overall (d 1–42)						
ADG (g/bird per day)	53.86	54.36	52.48	52.35	0.42	0.244
ADFI (g/bird per day)	92.08	91.85	89.45	91.08	0.87	0.725
FCR (g/g)	1.71	1.69	1.706	1.741	0.016	0.745
Mortality	4.76	3.57	2.38	3.57	1.09	0.908

Data are means of 7 replicate cages in each group. ADG = average daily gain; ADFI = average daily feed intake; FCR = feed conversion ratio.

**Table 5 animals-10-00225-t005:** Effects of fermented soybean meal (FSBM) on slaughter performance of broiler chickens.

Item	Dietary Treatment				SEM	*p*-Value
	Control	2.5% FSBM	5.0% FSBM	7.5% FSBM		
Final body weight (g)	2516	2475	2300	2382	41	0.259
Hot carcass weight (g)	1940	1915	1776	1824	33	0.249
Carcass yield (%)	77.23	77.38	77.23	76.51	0.15	0.156
Breast muscle (%)	31.11	30.4	31.19	31.37	0.23	0.488
Leg muscle (%)	20.07 ^a^	18.75 ^b^	18.73 ^b^	19.35 ^ab^	0.19	0.036
Abdominal fat (%)	1.36	1.48	1.32	1.46	0.06	0.773

^a,b^ Means in the same row, without the same superscript, differ significantly (*p* < 0.05). Data are means of 14 birds per group.

**Table 6 animals-10-00225-t006:** Effects of fermented soybean meal (FSBM) on meat quality traits of broiler chickens.

Item		Dietary Treatment				SEM	*p*-Value
		Control	2.5% FSBM	5.0% FSBM	7.5% FSBM		
Ultimate pH	Breast	5.53 ^b^	5.74 ^b^	5.56 ^b^	6.69 ^a^	0.09	<0.001
	Thigh	5.39 ^c^	5.94 ^b^	5.50 ^c^	6.60 ^a^	0.1	<0.001
Lightness (L*)	Breast	62.50 ^a^	51.50 ^b^	44.64 ^c^	53.79 ^b^	1.1	<0.001
	Thigh	64.29 ^a^	54.93 ^c^	54.14 ^c^	59.29 ^b^	0.83	<0.001
Redness (a*)	Breast	16.57 ^a^	15.21 ^a^	11.79 ^b^	16.86 ^a^	0.52	0.001
	Thigh	11	10.21	8.86	10.79	0.33	0.085
Yellowness (b*)	Breast	9.79 ^a^	8.29 ^a^	6.50 ^b^	8.43 ^a^	0.33	0.004
	Thigh	5.14	5.36	4.62	4.71	0.38	0.893
Shear force (N)	Breast	82	80.3	86.2	84.3	2.79	0.888
	Thigh	27.7 ^ab^	34.3 ^ab^	35.3 ^a^	26.1 ^b^	1.45	0.049
Drip loss (%)	Breast	2.61 ^a^	2.01 ^bc^	2.06 ^c^	2.57 ^ab^	0.1	0.034
	Thigh	1.77	1.44	1.6	1.51	0.06	0.32
Cooking loss (%)	Breast	17.16 ^a^	13.29 ^c^	14.42 ^bc^	16.01 ^ab^	0.4	0.002
	Thigh	12.97	12.24	12.63	11.88	0.55	0.914

^a–c^ Means in the same row without the same superscript differ significantly (*p* < 0.05). Data are means of 14 birds per group.

**Table 7 animals-10-00225-t007:** Effects of fermented soybean meal (FSBM) on muscle antioxidative status of broiler chickens ^1^.

Item ^2^		Dietary Treatment				SEM	*p*-Value
		Control	2.5% FSBM	5.0% FSBM	7.5% FSBM		
T-SOD (U/mgprot)	Breast	47.62 ^b^	55.62 ^a^	51.53 ^ab^	51.77 ^ab^	0.87	0.013
	Thigh	70.28	73.78	74.2	82.94	2.19	0.204
MDA (nmol/mgprot)	Breast	0.81 ^a^	0.61 ^b^	0.59 ^bc^	0.46 ^c^	0.03	<0.001
	Thigh	1.27	1.28	1.4	1.37	0.04	0.582
H_2_O_2_ (nmol/mgprot)	Breast	1.43 ^a^	1.34 ^a^	1.33 ^a^	1.00 ^b^	0.05	0.014
	Thigh	1.92 ^b^	2.34 ^ab^	2.13 ^b^	2.62 ^a^	0.09	0.022

^a–c^ Means in the same row without the same superscript differ significantly (*p* < 0.05). ^1^ Data are means of 14 birds per group. ^2^ T-SOD = total superoxide dismutase; MDA = malondialdehyde; H_2_O_2_ = hydrogen peroxide.

**Table 8 animals-10-00225-t008:** Effects of fermented soybean meal (FSBM) on the contents of free amino acids (μg/g wet tissue) in the muscle of broiler chickens ^1^.

Item		Dietary Treatment				SEM	*p*-Value
		Control	2.5% FSBM	5.0% FSBM	7.5% FSBM		
Sweet tasting amino acids							
Alanine	Breast	1081.1 ^b^	2090.8 ^a^	1144.9 ^b^	2259.1 ^a^	107.6	<0.001
	Thigh	2945.2	3613.4	3390	3089.2	131.8	0.282
Glutamine	Breast	351.7 ^a^	131.6 ^c^	262.4 ^b^	253.2 ^b^	18.83	<0.001
	Thigh	8183.9	7452.4	7412.2	6558.2	388	0.552
Glycine	Breast	1287.2 ^a^	1302.1 ^a^	595.3 ^b^	1435.3 ^a^	77.2	<0.001
	Thigh	1781.9	2226.9	2074.1	1772.4	96.1	0.258
Lysine	Breast	237.0 ^c^	616.4 ^a^	470.8 ^b^	454.5 ^b^	30.8	<0.001
	Thigh	1163	1390.4	1295.7	1199.7	74.9	0.727
Phenylalanine	Breast	173.3 ^b^	377.0 ^a^	182.6 ^b^	343.1 ^a^	19.6	<0.001
	Thigh	228.3	305.2	272.2	277.1	10.5	0.074
Serine	Breast	656.9 ^c^	1453.4 ^a^	759.6 ^c^	1032.2 ^b^	60.2	<0.001
	Thigh	1867.1 ^b^	2432.5 ^a^	1987.4 ^b^	1880.4 ^b^	79.1	0.028
Threonine	Breast	184.9 ^d^	865.1 ^a^	471.3 ^c^	637.2 ^b^	46.5	<0.001
	Thigh	823.5	959.2	946.8	840.1	31.7	0.303
Bitter tasting amino acids							
Arginine	Breast	262.4 ^d^	862.2 ^a^	405.3 ^c^	519.0 ^b^	43.9	<0.001
	Thigh	1888.6	1999	1825.8	1439.8	146.3	0.579
Histidine	Breast	51.4 ^d^	404.1 ^a^	101.0 ^c^	197.7 ^b^	25.1	<0.001
	Thigh	247.8	292.8	256.6	269.8	15.6	0.776
Isoleucine	Breast	96.8 ^d^	721.8 ^a^	176.6 ^c^	232.8 ^b^	44.5	<0.001
	Thigh	174.6 ^b^	214.6 ^ab^	240.8 ^a^	267.0 ^a^	11.7	0.021
Leucine	Breast	122.1 ^d^	622.9 ^a^	287.2 ^c^	426.6 ^b^	34.2	<0.001
	Thigh	303.6 ^b^	358.9 ^ab^	440.1 ^a^	448.0 ^a^	21.3	0.031
Methionine	Breast	79.8 ^d^	467.0 ^a^	343.5 ^c^	406.0 ^b^	27.6	<0.001
	Thigh	150.2 ^b^	195.0 ^ab^	212.1 ^ab^	244.3 ^a^	11.9	0.029
Valine	Breast	160.2 ^d^	489.2 ^a^	235.9 ^c^	330.6 ^b^	23.9	<0.001
	Thigh	273	306.6	348.6	373.2	14.7	0.058
Sour tasting amino acid							
Aspartic acid	Breast	62.7 ^c^	268.8 ^a^	165.0 ^b^	265.3 ^a^	16	<0.001
	Thigh	477.1	522.2	421.7	458.6	37.9	0.855
Umami tasting amino acid							
Glutamic acid	Breast	806.1 ^c^	1292.1 ^ab^	1078.8 ^bc^	1493.1 ^a^	69.1	0.001
	Thigh	1598.8	2065.6	1499.6	2019.9	118.3	0.219
Total free amino acids (mg/g wet tissue)	Breast	36.6 ^c^	49.3 ^a^	41.2 ^bc^	43.9 ^ab^	1.3	0.003
	Thigh	36.3	35.6	37.3	35.1	0.7	0.766

^a–d^ Means in the same row without the same superscript differ significantly (*p* < 0.05). ^1^ Data are means of 8 birds per group.

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
