# Peer review of "Partial Substitution of Fermented Soybean Meal for Soybean Meal Influences the Carcass Traits and Meat Quality of Broiler Chickens"

_animals, 2020, doi:10.3390/ani10020225_

Round 1

Reviewer 1 Report

Major comments:

Please proof-read your paper and ask for help from editing service. The manuscript has typos and grammar. Please write the manuscript in a scientific manner. Avoid uncertain words and exaggerated statements. The conclusion is not supported by results. The reviewer suggests running a regression analysis since treatments were increasing levels of fermented soybean-meal, 0, 2.5%, 5%, and 7.5%. Not all treatments affected measurements in a linear manner. There might be a quadratic regression relationship between the dietary FSBM addition and carcass quality and amino acid contents.

Simple Summary:

Line 21: duo?

Line 19: What does the “acceptability” mean? Please choose scientific vocabulary. 

Why do authors believe that the withdrawal of GPA increases the cost? Withdrawing drugs in feed directly decreased the cost of drug use. Please add reasoning and references.

Line 22: Do not use the abbreviation at the beginning of a sentence.

Line 24: Probiotics are direct-fed microbials. In my understanding, FSBM is a fermented soybean meal that contains an unfermented soybean meal, metabolites generated from fermentation, and microbials. It is not right to describe FSBM as a probiotic. Please revise it.

Delete the uncertain word, “some” in Line 27.

Line 28-30: Your data couldn’t support the statement. There was no statistical difference. Then, authors should not make this statement. Please correct it throughout the manuscript, in abstract, summary, results, discussion, and conclusion.

Introduction:

The introduction is weak and shallow. Please add specific information.

Please delete the duplicative sentence in Line 50 which had the same meaning as the previous sentence.

Line 51: Please correct the grammar. SBM decreased …factor?

In young animals? Please spell out the species.

Line 54: What do authors mean by “eliminate the most of trypsin inhibitors?” Will all fermentation processes eliminate inhibitors? What exactly the inhibitors were eliminated? The reviewer believes fermentation processes are temperature related and depend on strains of microbials. The authors only refereed to one yeast paper which was not enough.

Line 56 to 58, 60:  please list the exact improvements in growth performance, digestive enzyme activities, and morphology measurements.

Please add background information about the other two strains of your probiotics, Lactobacillus, and the yeast.

The authors studied carcass quality and nutrient contents. But no background information or the importance of the study was introduced.

Materials and Methods:

2.1

Line 78: Did broilers receive a 23-h light program throughout the whole grow-out trial? It is not approved by animal welfare regulation.

Line 80: Change the weird vocabulary, “adopt”.

How was the feed processed? Pelleted? Crumbled? Will the temperature affect the efficacy of the testing product? Did authors exam numbers probiotic strains in the finished feed or just in the fermented SBM?

Please add a reference to the broiler breed. Commercial brand and manufacture.

Table 1. How did the authors measure the probiotic counts? Did authors test all three category microbials, including B. subtilis, Lactobacillus spp. and yeast? What methods or agars did you use? Family names of bacteria and yeast should be italic. Correct them throughout the paper.

Please add the exact strain information of the yeast product and counts of each probiotic strain if available. The experiment must be repeatable based on the manuscript.

Please add a statement about the four dietary treatments were formulated with equal ME and protein levels.

2.6 Authors classified free amino acids by their taste. Please add the reference.

Results

Line 168: please correct the word, “in vivo”, not appropriate here.

Line 172, 203 to 205: P values were 0.244 and 0.725 for overall ADG and ADFI respectively, which meant no difference and not even a trend (P < 0.1). The authors couldn’t make the statement that 2.5% FBSM increased the ADG and FCR. It is unacceptable. Please correct the statement throughout the paper (summary, results, discussion, and conclusion). It damages the credibility of the paper and lowers the quality of your research!

Discussion

Line 263 to 270: Please move the whole paragraph to the introduction.

Line 271: Delete the emotional word, “undoubtedly”.

Please delete the sentence, “Research about the partial substitution of FSBM for SBM is limited”. Authors just claimed that FSBM showed improve growth performance from other studies.

Again remove the discussion about ADG and FCR among your treatments. Not correct.

Line 276: How would growth promote the effects of Bacillus help to explain that your product did not work? Confused.

Line 284: Why did the authors discuss genetic, hormonal, and nutritional factors, which were not studied in the current experiment? They distracted and confused readers. Please don’t overstate the negative results of your research. The unaffected performance resulted in unaffected carcass yield which made sense.

Line 292 to 297, 307 and 309: Please move them to the Introduction.

Again, spell out the acceptable pH range in Line 306.

Revise the paragraph from line 321 to 328. The major findings in your research were not discussed adequately. Authors did good research, please also spent time in developing your manuscript.

Delete “some” line 331.

Conclusion:

Revise the sentence “7.5% FSBM …greatest modulatory effects” in a scientific manner.

Delete the statement that "2.5% FSBM had superior growth performance". It was not supported by your data.

Reviewer 2 Report

It is necessary to provide the explanation of why free amino acid increases (by lysine, threonine, arginine, histidine, isoleucine, leucine, methionine, valine, aspartic acid) were observed in the breast muscles but in the thigh muscles was found only increase of serine (group with 2.5% FSBM), isoleucine and leucine (group with 5 and 7.5% FSBM) and methionine (group with 7.5% FSBM). Why are such differences in the  breast and thigh muscles? Why in the groups with dietary treatment of 2.5% and 7.5% FSBM  was found the increase of alanine, phenylalanine, serine, glutamic acids  in the breast muscles, but in the group with dietary treatment of 5 % FSBM, these differences were not significant in  comparison to control group. Could you explain the reason  why only content of glutamine (from all  measured amino acids) decreased in  comparison to control group.  In line 326 (page 15) it is written " FSBM treatments promoted the antioxidative properties of breast muscle".  How you can be explain that this was not found in the thigh muscles. Not suitable expression: "FSBM group had superior growth performance without statistical significance" (lines 346-347, page 16). Table 1 (page 4) is not in correct format.

Reviewer 3 Report

See attached PDF document

Round 2

Reviewer 2 Report

Thank you for accepting my notes and correcting according them the manuscript. Now, in my opinion, the quality of the manuscript has greatly improved and it can be accepted for printing in the journal.

Reviewer 3 Report

Authors have made concerted efforts to improve this manuscript based on reviewers' suggestions. I recommend publication in Animals.